# *Crypthecodinium cohnii* Growth and Omega Fatty Acid Production in Mediums Supplemented with Extract from Recycled Biomass

**DOI:** 10.3390/md20010068

**Published:** 2022-01-12

**Authors:** Elina Didrihsone, Konstantins Dubencovs, Mara Grube, Karlis Shvirksts, Anastasija Suleiko, Arturs Suleiko, Juris Vanags

**Affiliations:** 1Latvian State Institute of Wood Chemistry, LV1006 Riga, Latvia; gmtd@inbox.lv (K.D.); anastasija.gurcinska@gmail.com (A.S.); arturs.suleiko@bioreactors.net (A.S.); btc@edi.lv (J.V.); 2A/S Biotehniskais Centrs, LV1006 Riga, Latvia; 3Institute of General Chemical Engineering, Faculty of Materials Science and Applied Chemistry, Riga Technical University, LV1048 Riga, Latvia; 4Institute of Microbiology and Biotechnology, University of Latvia, LV1004 Riga, Latvia; mara.grube@lu.lv (M.G.); karlis.svirksts@lu.lv (K.S.)

**Keywords:** *Crypthecodinium cohnii*, omega-3 fatty acid, biomass recycling, dinoflagellate extract, FTIR spectroscopy

## Abstract

*Crypthecodinium cohnii* is a marine heterotrophic dinoflagellate that can accumulate high amounts of omega-3 polyunsaturated fatty acids (PUFAs), and thus has the potential to replace conventional PUFAs production with eco-friendlier technology. So far, *C. cohnii* cultivation has been mainly carried out with the use of yeast extract (YE) as a nitrogen source. In the present study, alternative carbon and nitrogen sources were studied: the extraction ethanol (EE), remaining after lipid extraction, as a carbon source, and dinoflagellate extract (DE) from recycled algae biomass *C. cohnii* as a source of carbon, nitrogen, and vitamins. In mediums with glucose and DE, the highest specific biomass growth rate reached a maximum of 1.012 h^−1^, while the biomass yield from substrate reached 0.601 g·g^−1^. EE as the carbon source, in comparison to pure ethanol, showed good results in terms of stimulating the biomass growth rate (an 18.5% increase in specific biomass growth rate was observed). DE supplement to the EE-based mediums promoted both the biomass growth (the specific growth rate reached 0.701 h^−1^) and yield from the substrate (0.234 g·g^−1^). The FTIR spectroscopy data showed that mediums supplemented with EE or DE promoted the accumulation of PUFAs/docosahexaenoic acid (DHA), when compared to mediums containing glucose and commercial YE.

## 1. Introduction

One of the most commercially important representatives of the omega-3 fatty acids’ (FAs) group is docosahexaenoic acid (DHA), which is a long-chain, highly polyunsaturated omega-3 (n-3) fatty acid (LC-PUFA). DHA is considered one of the most significant and beneficial fatty acids for the health of infants and adults. Numerous research papers have reported that DHA supports the human cardiovascular and nervous systems, prevents the occurrence of inflammatory diseases, alleviates depression, and treats psoriasis and rheumatoid arthritis. Furthermore, DHA plays a key role in the healthy development of the fetal brain and retina, thus it is commonly included in infant-oriented food products and supplements [1,2].

Currently, the main source of DHA is fish oil. However, when compared to microbial DHA, fish-derived PUFAs lack the flexibility of its biosynthetic counterpart, as availability of raw materials (e.g., fish oil for its production) strongly depends on fish resources (e.g., seasonality and geographical location). The fish oil purification process is also quite difficult, and the resulting product remains unsuitable for all dietary requirements (e.g., vegetarians and vegans) [3,4,5]. Furthermore, fish oil has a specific odour and taste, which is unpleasant for a noticeable part of people, especially infants. Therefore, omega-3 FA obtainment from fish is suboptimal and poses a negative effect on the environment. Moreover, conventional DHA production currently cannot meet the increasing demand for omega-3 FA for human consumption [5].

A wide misconception is that fish produce DHA themselves through specific metabolic pathways, which are semi-unique to aquatic life forms. Marine organisms, especially different families of fish, in their natural habitats accumulate omega-3 FAs in their organisms through feeding on zooplankton, which in turn consumes the primary omega-3 FAs producers, namely microalgae [2]. However, in fish farms, the use of eicosapentaenoic acid (EPA) and DHA as feed supplements has become a conventional practice. The demand for products containing omega-3 FA has significantly increased during the past decades. However, due to insufficient fish resources, the global market currently is in crisis. As of now, the global fish oil production reaches approximately 1 million metric tonnes per year, of which ~70% is generally used for aquafeeds [6].

Considering all of the above mentioned, direct methods for FAs acquisition from unicellular microorganisms, which have the ability to synthesize DHA on their own, becomes preferable, even though the cost of edible microbial oil is estimated to reach 3000–5000 USD per kg [7], which is considerably higher than conventional fish oil, estimated to exceed 2000 USD per metric ton [6]. Besides, the concentration of DHA in single cell oil (SCO), for example, from *Crypthecodinium cohnii*, can reach much higher concentrations when compared to fish oil (54% and 12% by mass, respectively) [8,9].

Cultivated microalgae-derived oil does not contain heavy metals and cholesterol, and has a neutral taste, which can be easily enhanced depending on the consumer requirements [2]. The average lipid content in microalgae biomass is from 20 to 50% by mass. However, under stress conditions, it can reach even higher levels (up to 85%) [10]. Microalgae species such as *C. cohnii*, *Nannochloropsis gaditana, Isochrysis galbana* and *Phaeodactylum tricornutum* were proven to be suitable for production of PUFAs on a commercial scale [10]. Multiple commercial scale applications were already previously studied and successfully put into commission (e.g., microalgae cultivation in tubular and flat panel bioreactors [10] and transgenic oilseed plants [11]), which indicates the severity of the DHA shortage that the world is experiencing right now.

A marine dinoflagellate *C. cohnii* can accumulate PUFAs in significant amounts (up to 25% of DHA or 35% of FAs of dry weight [2,8] )and therefore it has been used previously for industrial production of omega-3 fatty acids [12]. However, in *C. cohnii* cultivation processes, yeast extract (YE) is conventionally used as the nitrogen source, which noticeably affects the cost of the target products [3]. Therefore, the identification of a cheap and renewable substrate for a highly efficient DHA production by *C. cohnii* is necessary. In the literature, suitable carbon sources have been widely studied (e.g., glucose, acetate, glycerol, oleic acid, acetic acid, ethanol, rapeseed meal hydrolysate, crude waste molasses, cheese whey, corn steep liquor, tagatose, carob syrup, date syrup, and galacturonic acid) [3,4,13,14,15,16]. The results (see Table 1) show that the highest biomass concentrations were achieved using acetic acid and ethanol in fed-batch fermentations (109 g·L^−1^ and 83 g·L^−1^, respectively [17,18]) and in batch fermentations with glucose and acetate (27.7 g·L^−1^ and 7.03 g·L^−1^, respectively [4,19]). The highest DHA concentrations 19 g·L^−1^ and 11.7 g·L^−1^ were achieved using acetic acid and ethanol, respectively, as carbon sources in fed-batch fermentations [17,18]. In batch fermentations, the highest DHA titres have been achieved with glucose (1.6 g·L^−1^ and 1.4 g·L^−1^) [19,20]. However, the effect of nitrogen sources on the cultivation efficiency has not been studied as extensively as carbon sources. Suitable nitrogen sources for marine protists are tryptone, yeast extract, peptone, soy peptone, urea, monosodium glutamate, nitrate, ammonia, and ammonium chloride [3,13]. It also should be noted that some marine protists (e.g., *Schizochytrium* species) can utilize a wider range of nitrogen sources than others (e.g., *Crypthecodinium* species) [13]. Although nitrogen source variation in *C. cohnii* cultivations has been employed, e.g., urea, yeast extract, meat extract, glutamic acid, ammonium sulphate, ammonium bicarbonate, sodium nitrite, and ammonium nitrate [15,21,22], yet mostly the effect of the nitrogen source on the *C. cohnii* growth has not been the focus of the past studies. The highest DHA titres in microalgae cells—0.99 g·L^−1^, has been observed when sodium nitrate was utilized [22]. The highest total lipid content, 28.48%, 18.67%, and 18.14% of dry cell weight (DCW), has been observed utilizing threonine, yeast extract, and sodium nitrate, respectively [21,22,23]. Moreover, to the authors’ knowledge, there have been no attempts to use a recycled waste product as a nitrogen source as it will be outlined in the present study.

An alternative way of cultivation could be to use the extraction ethanol (EE), remaining after lipid extraction, as a source of carbon, and extracts from recycled dinoflagellate biomass as a source of carbon, nitrogen, and vitamins. A substitute of conventional nitrogen sources, dinoflagellate extract (DE), is obtained from de-oiled microalgae biomass (i.e., after lipid extraction, by hydrolysis, neutralization with calcium carbonate, sedimentation, separation, evaporation of liquid phase, and drying). The described process can also be called biomass recycling.

The aim of the present study was to evaluate the growth and metabolic response of *C. cohnii* to different carbon and nitrogen sources in growth media including conventional commercially available YE, and two novel extracts (EE, Des). This approach provides more efficient use of the lipid extraction by-products/waste products and circular DHA production process (Figure 1), and therefore could be beneficial to the bio-economy.

## 2. Results

To access the possibility of replacing YE with cheaper alternative sources of nitrogen and nutrients for the cultivation of *C. cohnii* and DHA production, we used extracts obtained from de-oiled dinoflagellate biomass, as well as ethanol, which was used as part of one of the oil extraction methods. Two main methods were used to extract oil from the dinoflagellate biomass (see Figure 2). In the first case, the oil was extracted from lyophilized biomass using hexane. Thus, obtaining a SCO, which after esterification can be separated into FAs. A waste product of this process is de-oiled microalgae biomass, which is hydrolysed to obtain dinoflagellate extract (DE) (see Figure 2A). In the second case, the oil extraction is carried out through saponification of fats in wet biomass with KOH in the presence of ethanol. Thus, obtaining the hydroalcoholic phase, containing soaps, and de-oiled microalgae biomass. Ethanol, which is used in this process, extracts multiple components from the biomass and can serve as a source of carbon and organic nitrogen, vitamins, nutrients, and salts for subsequent cultivations (see Figure 2B).

The first method requires additional biomass processing before the oil extraction (e.g., freezing and drying (lyophilization). On the contrary, the second method requires an additional extraction step. The obtained DEs by the first and second method were called DE_A_ and DE_B_, respectively. Experiments on the effect of the DE on the growth of *C. cohnii* were carried out in mediums with glucose as the main carbon source. EE was used as an alternative source of carbon.

### 2.1. Experiments with Glucose as the Carbon Source

Multiple cultivation experiments were carried out using a complex medium containing glucose, sea salt, and YE and/or DE to study the effect of DE on the growth rate, biomass, and lipids including DHA by *C. cohnii*. The media composition used in the experiments with glucose as the main carbon source are summarized in Table 2. The mediums under study contained either only YE, only DE, or YE and DE (25/75 *w*/*w*). Selection of the initial glucose concentration (10 g·L^−1^) is justified by the fact that the mentioned amount of substrate is enough for the biomass to fully consume 1 g·L^−1^ of YE. Furthermore, de Swaaf et al. [17], Jiang et al. [26], and Diao et al. [27], as part of previously reported studies, have shown that the maximum biomass growth rate is achieved if the glucose concentration is maintained in the range of 5–25 g·L^−1^. Additionally, de Swaaf et al. has demonstrated, that biomass growth inhibition begins at glucose concentrations of 20–25 g·L^−1^. The specific biomass growth rates and yields in different mediums are shown in Table 2. The maximum specific biomass growth rate and yield from glucose were observed in the medium containing exclusively DE_A_, and were equal to 1.012 and 0.601 g·g^−1^, respectively. The lowest specific growth rate (0.655 h^−1^) was observed with the medium, which contained only DE_B_. A similar growth rate (0.615 h^−1^) was observed with the reference medium with no added extracts.

The addition of 25% YE to 75% DE_B_ into the cultivation medium (DE_B75_) increased the specific biomass growth rate by 37% (up to 0.901 h^−1^), while the yield of biomass remained mostly unchanged (0.398 g·g^−1^). However, the addition of 25% of YE to DE_A_ (medium DE_A75_) lowered the specific biomass growth rate to 0.9 h^−1^, in comparison to the mediums containing only DE extracts.

Figure 1 shows the biomass growth (A) and glucose consumption (B) curves in mediums given in Table 2. It can be seen that all mediums containing YE ensured complete assimilation of glucose in 7–14 days, while in mediums containing DE, only half of the initially supplemented substrate was assimilated until the 14th experiment day.

Almost all growth curves, except the DE_A_ medium, reached the lag phase during the first day of cultivation. The medium containing only YE showed the highest average specific growth rate for the first three days until all glucose was consumed. After that, the growth passed into the stationary phase and cyst formation began. In other mediums, the maximum specific growth rate was observed only on the first cultivation day, after which the biomass growth remained constant.

### 2.2. Experiments with Ethanol as the Carbon Source

The EE, collected after the lipid extraction from *C. cohnii* biomass, was used as the main carbon source in complex mediums containing YE and DE as sources of nitrogen and nutrients. The media compositions, the specific biomass growth rates, and biomass yields are summarized in Table 3.

The highest specific growth rate and biomass yield were obtained in mediums containing YE and reached 0.757 h^−1^ and 0.282 g·g^−1^, respectively. In mediums with DE, the growth rates were slightly lower than using YE and reached 0.701 h^−1^ for DE_A_ and 0.651 h^−1^ for DE_B_. The biomass yield on the 14th day of cultivation in mediums containing DE_A_ and DE_B_, reached 0.234 and 0.221 g·g^−1^, respectively.

From Figure 2 it can be observed that in the case of YE and DE_A_, the specific biomass growth rate reached the maximum and remained constant until the 4th cultivation day until the substrate was not entirely consumed. In turn, the biomass growth rate in mediums containing only DE_B_ was relatively high only on the first day of cultivation, after which it gradually decreased. It should be noted that during cultivation in mediums containing YE, similarly as in the glucose experiment, the lag phase was observed during the first day of cultivation.

Experiments on pure and extraction ethanol (EE) were conducted to evaluate their effect on the biomass growth rate and yield. The specific growth rate in the case of EE was 0.470 h^−1^, which is for 20% more than in pure ethanol (0.383 h^−1^) and 50% more than in mediums containing YE or DE. The maximum biomass yield in both cases was very similar (0.124 g·g^−1^ for EE and 0.122 g·g^−1^ for pure ethanol), but with EE it was reached on the seventh day, and with pure ethanol on the 10th cultivation day.

### 2.3. Evaluation of Lipid/FA and PUFA Accumulation in C. cohnii Biomass by FTIR

FTIR is a rapid method, particularly used for monitoring the relative content of each macromolecular component under varying growth conditions [28,29,30,31]. FTIR spectroscopy of *C. cohnii* biomass was used to evaluate the growth medium-induced production of lipids/FA and PUFAs. Fish oil supplements naturally contain about 30% of EPA and DHA in the form of triacylglycerols (TAGs), a tri-ester [32,33]. The FTIR spectrum of fish oil (Figure 3) reveals three high-intensity absorption bands at 2925, 2854 cm^−1^ (CH_3_ and CH_2_ vibrations, respectively), and 1745 cm^−1^ (C=O vibrations of lipid esters) that are indicative of lipids, FAs or triglycerides and therefore are indicative of total lipids. The spectrum also revealed a smaller peak at 3011 cm^−1^ (olefinic group = CH), which is typical for unsaturated fatty acids (PUFAs/DHA) [34,35,36,37,38,39]. 

The characteristic absorption bands of the major cell components in the FTIR spectra are at 1080 cm^−1^ of carbohydrates; 1250 cm^−1^ of nucleic acids; 1650 and 1545 cm^−1^ of proteins (Amide I and Amide II, stretching vibrations of C=O bond of amide and bending vibrations of the N-H bond, respectively); triplet bands in 2800–3000 cm^−1^ and 1744 cm^−1^ of lipids/FA (C-H stretching in CH_3_ and CH_2_ and C=O of esters/ester carbonyl, respectively) and ~3014 cm^−1^ of PUFAs/DHA (olefinic HC=CH stretching mode) [28,40]. The position and intensities of particular absorption bands allow to monitor or evaluate the macromolecular composition of cells as well as the accumulation of lipids/FAs and PUFAs [40,41,42,43]. PUFAs in the FTIR spectrum show a peak in the range of 3005–3013 cm^−1^, particularly the specific peak of DHA oils is at ~3013,4 cm^−1^ [44].

Samples for FTIR were collected only on day 14, due to the amount of accumulated biomass in the experimental setup. For data analysis of *C. cohnii* cells, only spectra with absorption limits between 0.25 and 0.80 were used, and therefore, in accordance with the Lambert-Bouger-Beer law, the concentration of a component is proportional to the intensity of the absorption band. The spectra were vector normalized and therefore the intensity of the vibration band was proportional to the amount of band vibrations, i.e., the intensity is proportional to concentration. Therefore, the latter allows to cross-compare the cell biomass composition, accumulation, and number of macromolecular components (e.g., proteins, carbohydrates, FAs, PUFAs, DHA, etc). This is an especially valuable FTIR spectroscopy approach for quick and informative evaluation of large sample sets to select the best growth conditions for the production/accumulation of the targeted metabolites. Further quantitative and qualitative analyses of the most relevant samples can be carried out more precisely by FTIR spectroscopy, chromatography, mass spectroscopy, etc. Therefore, even though FTIR is a semi-quantitative method and does not provide precise values, it remarkably saves resources and time for evaluation of different biotechnological processes. 

FTIR spectra of *C. cohnii* grown in mediums with YE, DE_A_, DE_A75_, DE_B75_, or glucose (Figure 4) showed that the macromolecular composition of cells is different depending on the growth medium composition. The spectrum profile of the *C. cohnii* cells grown in medium containing YE was noticeably different from others of this experimental set. Spectra of the cells grown with YE showed similar amounts of total carbohydrates but higher content of proteins and lower content of total lipids than in cells grown in mediums with DE_A_, DE_A75_ DE_B75_, or glucose. The vector normalized spectra of cells grown without YE showed similar content of the total carbohydrates and proteins, but the content of lipids/FA and PUFAs/DHA varied. The highest number of total lipids/FAs (2925, 2854, and 1745 cm^−1^) was detected in cells grown in mediums with glucose but lower with DE_A75_ and DE_B75_. However, a higher amount of PUFAs/DHA (3014 cm^−1^) was detected in cells grown in medium with DE_A75_.

FTIR spectra of *C. cohnii* cultivated in mediums with EE-YE, EE-DE_A_, EE-DE_B_, EE-DE_A75_, EE-DE_B75_, EE, or pure ethanol are shown in Figure 5. FTIR spectra showed different cell macromolecular compositions, which can be grouped into two clusters. The first group EE-YE, EE-DE_A75_, and EE-DE_B75_ produce relatively high amounts of proteins, low amounts of total carbohydrates, and lesser amounts of total lipids than cells grown with EE-DE_A_, EE-DE_B_, EE, or pure ethanol. The FTIR spectra of the second group (i.e., cells grown in EE-DE_A_, EE-DE_B_, EE, or pure ethanol showed more total lipids/FA and PUFAs/DHA compared to those of the first group). The highest content of FAs and PUFAs/DHA was detected in *C. cohnii* grown in mediums with EE/EE-DE_A_ and EE-DE_B_.

## 3. Discussion

The main components of the culture medium for *C. cohnii* are the substrate (glucose, glycerol, ethanol, acetic acid), yeast extract, and sea salt (artificial or natural). The biomass yield from each component of the cultivation medium was determined during bioreactor cultivations and reached 0.7 g·g^−1^ for glucose, 5.3 g·g^−1^ for yeast extract, and 0.87 g·g^−1^ for sea salt (results not shown). Despite the fact that, compared to other components, a small amount (in terms of mass) of yeast extract is used, its price is the highest (8–10 USD/kg) and makes up more than half of the cultivation medium cost. Considering that the costs of raw materials for the production of SCO is about half of all production expenses, the reduction of the necessary amounts of cultivation medium components poses a significant effect on the overall process economy.

The obtained results of this study show that DE_A_ stimulated a similar biomass growth rate when compared to YE, while the biomass yield was significantly higher if glucose was used as a carbon source. The absence of a lag phase during cultivation in mediums containing only DE_A_, both with glucose or ethanol, unlike mediums with YE, indicates that the mineral and vitamin composition of DE_A_ is more preferable for proliferation of *C. cohnii* cells. However, a decrease of the biomass growth rate in mediums with DE_A_ should be noted. The increase in biomass titres after cultivation for four days (media with DE_A_) and seven days (media with DE_A75_) becomes constant and equal to that obtained in the control medium (with glucose). Therefore, despite the high initial biomass growth rates, the efficiency of DE_A_ is approximately two times lower when compared to that of YE. The latter can be compensated by increasing the initial concentrations of DE_A_ or the addition of small amounts of YE to the cultivation medium, which in itself depends on the economic feasibility of the process.

The use of DE_B_ as a nitrogen source, in the case of both glucose and ethanol, had a minimal effect on the growth rate and biomass yield. Obviously, many thermal and chemical biomass processing steps, which are necessary to obtain DE_B_, completely or to a larger extent degrade the initial vitamins present in *C. cohnii* biomass.

EE, although it did not significantly affect the biomass growth rate, showed the same biomass yield when compared to the control sample (pure ethanol), which indicates the feasibility of using it as a carbon source for *C. cohnii* cultivation.

Cross comparison of the FTIR spectral data showed that the growth medium components clearly affect the biochemical composition of *C. cohnii* cells. When grown in mediums with YE, EE-YE, EE-DE_A75_, or EE-DE_B75_, *C. cohnii* cells contained more proteins (compared to the cells grown in any other studied mediums) but the absorption band at 1745 cm^−1^ (ester C=O bonds of lipids/FA) was not detected, thus indicating that the cells did not overproduce lipids/FA. In the context of searching for the growth conditions promoting the accumulation of PUFAs/DHA the most promising results were acquired when *C. cohnii* was cultivated in the mediums with DE_A75_ and glucose (Figure 4), EE-DE_A_, EE-DE_B_, EE, or ethanol (Figure 5). However, further quantitative analyses of the main cell macromolecular components (carbohydrates, proteins, and lipids) are needed to identify the most efficient growth media that promotes the overproduction of PUFAs/DHA by *C. cohnii*. 

To summarize the above mentioned, DE_A_ and EE can both be successfully used as alternative sources of nitrogen, nutrients, and carbon to reduce the costs of conventional SCO and DHA production processes.

Moreover, the production of DE_A_ can be optimized through the use of hydrochloric acid and sodium hydroxide. The hydrolysate obtained in this way, after neutralization with acid, can be directly used to create a suitable dinoflagellate cultivation medium. Furthermore, the use of the above-mentioned hydrolysate will make it possible to exclude such energy-demanding steps of the production process as evaporation and simultaneously significantly reduce the amount of sea salt, which otherwise should be added in the cultivation medium in large quantities.

## 4. Materials and Methods

### 4.1. Cultivation Conditions

*Crypthecodinium cohnii* CCMP 316 was obtained from the Provasoli-Guillard National Center for Marine Algae and Microbiota (NCMA) (USA). The culture inoculum was grown in a custom-made setup (see Figure 3) consisting of 250 mL bottles with a working volume of 150 mL. The growth medium containing glucose 5 g·L^−1^, yeast extract 2 g·L^−1^ and sea salts 25 g·L^−1^, with aeration 30 mL/min, rotation speed 130 rpm (provided by an orbital shaker PSU-20i, Biosan, Riga, Latvia) was maintained at 25 °C. The cultivation medium compositions for experiments are shown in Table 4, as well as in Table 2 and Table 3. The initial optical density (OD_470_) of experiments was set to 0.15. 

### 4.2. Dinoflagellate Extract and Extraction Ethanol Obtainment

Dinoflagellate extracts (DEs) were obtained by recycling de-oiled *C. cohnii* biomass from the cultivation process in a laboratory scale bioreactor EDF-5.4_1 (JSC Biotehniskais centrs, Riga, Latvia) by a method adapted from Gao et al. [45]. The de-oiled biomass was hydrolysed with H_2_SO_4_ at 121 °C for 20 min. The hydrolysed solution was neutralized with an appropriate amount of CaCO_3_. The solid fraction was separated from the hydrolysate by filtration. Most of the liquid hydrolysate was evaporated by heating at 120 °C, 600 rpm. The remaining moisture was evaporated by drying the sample in an oven at 80 °C overnight. The obtained dry extract pellets were grinned into a fine powder, hereinafter referred to as DE. De-oiled biomass for DE_A_ obtainment was produced by a method adapted from Halim et al. [46]. Lipids were extracted from lyophilized biomass with the use of Soxhlet extraction with hexane as the solvent, containing 0.01% butylated hydroxytoluene (BHT). De-oiled biomass for DE_B_ obtainment was produced by a method adapted from Mendes et al. [47]. Lipids were extracted with hexane (BHT concentration in hexane 0.01%) from the wet biomass, followed by incubation overnight at 20 °C with ethanol and KOH, simultaneously the extraction ethanol (EE) was obtained and separated. The acidification with HCl and additional hexane extractions were performed, the resulting suspension was used for DE obtainment as described above.

### 4.3. Optical Density and Glucose Concentration Measurements

*C. cohnii* growth was monitored via optical density (OD) measurements at a wavelength of 470 nm with a spectrophotometer Jenway 6300 (Cole-Parmer, Saint Neots, UK). The glucose concentration was measured enzymatically with an AccuChek ACTIVE blood sugar analyzer (Roche, Basel, Switzerland). Samples for OD and glucose measurements were taken on days 1, 2, 4, 7, 10, and 14. The experiments were performed in at-least triplicate and the experimental data was expressed as mean ± standard deviation (SD).

### 4.4. Determination of Biomass Dry Cell Weight

The biomass dry cell weight (DCW) in relation to the absorbance at a wavelength of 470 nm was determined gravimetrically as described in [48]. During the present study, the correlation coefficients value was determined as 1.415 g (DCW)·L^−1^·A.U.^−1^.

### 4.5. Fourier Transform Infrared Spectroscopy

Fourier transform infrared (FTIR) spectra of biomass were recorded using Vertex 70 coupled with the HTS XT microplate reader (Bruker Optik GmbH, Ettlingen, Germany) Sample aliquots were pipetted on a 384 well microplate dried and recorded in the frequency range of 4000–600 cm^−1^. Omega fatty acids were identified by absorption bands at ~1743 cm^−1^ and ~3012 cm^−1^. Due to the experimental setup and amount of accumulated biomass during the cultivation processes, samples for FTIR were collected only on day 14.

For data analyses only spectra with absorption limits between 0.25 and 0.80 were used, and therefore, in accordance with the Lambert-Bouger-Beer law, the concentration of a component is proportional to the intensity of the absorption band. Spectra were vector normalized, and therefore the intensity of the vibration band was proportional to the amount of band vibrations (i.e., the intensity is proportional to concentration).

## 5. Conclusions

FTIR spectra of *C. cohnii* cells clearly showed the medium-induced metabolic responses, including variations of the produced total lipids/FA, PUFAs, and DHA. Further studies of the concentrations and composition of PUFAs produced under various cultivation conditions together with the cell growth data would allow us to identify the most efficient cultivation medium composition. Nevertheless, current observations point out to the positive effect (both in terms of process economy and process efficiency) of supplementing the standard *C. cohnii* cultivation medium with recycled components (e.g., dinoflagellate extracts and extraction ethanol).

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
