# Peer review of "Crypthecodinium cohnii Growth and Omega Fatty Acid Production in Mediums Supplemented with Extract from Recycled Biomass"

_marinedrugs, 2022, doi:10.3390/md20010068_

Round 1
Reviewer 1 Report
In the study, the authors investigated the effect of the extract ethanol and dinoflagellate extract on cultivation of Cryothecodinium cohnii, which can produce large amount of docosahexaenoic acid. Furthermore, the authors evaluated lipid accumulation in the microalgae by FTIR. These researches are necessary for establishment of industrial DHA production using the microalgae. However, I think there are some points to be revised and reconsidered.
Major points
- Abstract: The abstract section seems like a part of introduction section. The authors should mention about summaries of study backbone, results, and discussion.
- Table 2: Why did the authors decide to use 10g/L glucose. Please describe the reason with reference or something showing that the concentration is better.
- Fig 1, Fig 2, Table 2, and Table 3: At those time, how much fatty acids and DHA did C. cohnii produce? I think it can not be said that higher biomass leads higher functional fatty acids. The authors should compare them such as amount of total fatty acid, ratio of each fatty acid, and amount and ratio of DHA, for each cultivation. Moreover, the authors should show fatty acid composition in C. cohnii.
Minor points
- Page 2, line 60: The “C. cohnii” appear firstly at this location. Thus, please write full genus name here. At many lines, please change “C. cohnii” in italic.
- At many pages, many lines: Schema => Scheme. Please confirm them.
- Scheme 1: The authors should describe abbreviations of DE and SCO in the caption. Please unify description method e.g. Fermentation Media etc and Biomass filtration etc.
Author Response
Response to Reviewer 1 Comments
English language and style (x) Moderate English changes required
Is the research design appropriate? (x) Must be improved
Are the results clearly presented? (x) Can be improved
Response 1: Language changes has been introduced in the manuscript text. Research design and result presentation is improved according to the suggestions of Reviewer 1 and Reviewer 2.
Major points
- Abstract: The abstract section seems like a part of introduction section. The authors should mention about summaries of study backbone, results, and discussion.
Response 1: The abstract has been revised and the main results are included.
- Table 2: Why did the authors decide to use 10g/L glucose. Please describe the reason with reference or something showing that the concentration is better.
Response 2: The explanation of the selected glucose concentration is enclosed in the manuscript section 2.1. and copied below:
Selection of the initial glucose concentration (10 gꞏL-1) is justified by the fact that the mentioned amount of substrate is enough for the biomass to fully consume 1 gꞏL-1 of YE. Furthermore, de Swaaf et al.[1], Jiang et al. [2], and Diao et al.[3], as part of previously reported studies, have shown that the maximum biomass growth rate is achieved if the glucose concentration is maintained in the range of 5 – 25 gꞏL-1. Additionally, de Swaaf et al. has demonstrated, that biomass growth inhibition begins at glucose concentrations of 20 – 25 gꞏL-1.
- Fig 1, Fig 2, Table 2, and Table 3: At those time, how much fatty acids and DHA did C. cohnii produce? I think it can not be said that higher biomass leads higher functional fatty acids. The authors should compare them such as amount of total fatty acid, ratio of each fatty acid, and amount and ratio of DHA, for each cultivation. Moreover, the authors should show fatty acid composition in C. cohnii.
Response 3: The answer and comments regarding the fatty acid and DHA production by C. cohnii is enclosed in the manuscript section 2.3. and copied below:
Comment: In this study FTIR spectroscopy was used to cross-compare the composition of C. cohnii biomass and identify the growth conditions that provide the accummulation of lipids and PUFAs by a semi-quantitative approach, that allows to assess the metabolism, but does not give precise quantitative data.
Samples for FTIR spectroscopy were collected only on day 14th (see Materials and Methods).
Spectra were vector normalized and therefore the intensity of the vibration band was proportional to the amount of band vibrations, i.e., the intensity is proportional to concentration. Therefore, it allows to cross-compare the cell biomass composition, accumulation, and amount of macromolecular components e. g., proteins, carbohydrates, FAs, PUFAs, DHA, etc. This is an especially valuable and estimated FTIR spectroscopy approach for quick and informative evaluation of large sample sets to select the best growth conditions for the production/accumulation of the targeted metabolites. Further quantitative and qualitative analyses of the most relevant samples can be analysed more precisely by FTIR spectroscopy, chromatography, mass spectroscopy, etc. Therefore, even that FTIR semi-quantitative method does not give precise numbers it remarkably saves resources and time for evaluation of different biotechnological processes.
Minor points
- Page 2, line 60: The “C. cohnii” appear firstly at this location. Thus, please write full genus name here. At many lines, please change “C. cohnii” in italic.
Response 1: In page 2, line 60 full genus of the species have been included. C. cohnii in all lines are formatted to italic.
- At many pages, many lines: Schema => Scheme. Please confirm them.
Response 2: The term (Scheme) has been chosen and correected.
- Scheme 1: The authors should describe abbreviations of DE and SCO in the caption. Please unify description method e.g. Fermentation Media etc. and Biomass filtration etc.
Response 3: In Scheme 1 the abbreviations of DE and SCO are described in the caption. Unified style is introduced into the steps of the scheme.
Reviewer 2 Report
In this manuscript, the authors showed the effect of adding extractive ethanol and flagellum extract to the medium on changes in biomass growth rate, lipids and biomass production.
The content is good written, but the grammar needs to be further improvement, especially the abstract.
There are minor linguistic imperfections that can be easily corrected by hat authors after careful reading.
In the Methods section, include a graphical timeline or flow diagram of the study design/experimental procedure and provide a reference to the schematic in the manuscript text.
Specifies the statistical method used to predetermine the culture (optical density, glucose concentration) in the manuscript.
Author Response
Response to Reviewer 2 Comments
English language and style (x) Moderate English changes required
Are the methods adequately described? (x) Can be improved
Response 1: Language of the manuscript has been revised. Methods are more precisely described according to the reviewer’s suggestions. The Abstract is improved and revised.
The content is good written, but the grammar needs to be further improvement, especially the abstract.
There are minor linguistic imperfections that can be easily corrected by hat authors after careful reading.
Response 2: Language has been revised.
In the Methods section, include a graphical timeline or flow diagram of the study design/experimental procedure and provide a reference to the schematic in the manuscript text.
Response 3: Materials and Methods section 4.1. is revised, the Table 4. shows of all investigated media compositions.
Specifies the statistical method used to predetermine the culture (optical density, glucose concentration) in the manuscript.
Response 4: The statistical method used to predetermine the culture is specified in the manuscript section 4.3. and copied below:
The experiments were performed in at-least triplicate and the experimental data was expressed as mean ± standard deviation (SD).
Round 2
Reviewer 1 Report
The authors addressed and revised well.